# Enhanced generation and anisotropic Coulomb scattering of hot electrons in an ultra-broadband plasmonic nanopatch metasurface

Matthew E. Sykes[1], Jon W. Stewart [2], Gleb M. Akselrod[2], Xiang-Tian Kong [3,4], Zhiming Wang[3], David J. Gosztola[1], Alex B.F. Martinson [5], Daniel Rosenmann[1], Maiken H. Mikkelsen [2,6], Alexander O. Govorov [4] & Gary P. Wiederrecht[1]

The creation of energetic electrons through plasmon excitation of nanostructures before thermalization has been proposed for a wide number of applications in optical energy conversion and ultrafast nanophotonics. However, the use of "nonthermal" electrons is primarily limited by both a low generation efficiency and their ultrafast decay. We report experimental and theoretical results on the use of broadband plasmonic nanopatch metasurfaces comprising a gold substrate coupled to silver nanocubes that produce large concentrations of hot electrons, which we measure using transient absorption spectroscopy. We find evidence for three subpopulations of nonthermal carriers, which we propose arise from anisotropic electron–electron scattering within *sp*-bands near the Fermi surface. The bimetallic character of the metasurface strongly impacts the physics, with dissipation occurring primarily in the gold, whereas the quantum process of hot electron generation takes place in both components. Our calculations show that the choice of geometry and materials is crucial for producing strong ultrafast nonthermal electron components.

[1] Center for Nanoscale Materials, Argonne National Laboratory, Argonne, IL 60439, USA. [2] Department of Electrical and Computer Engineering, Duke University, Durham, NC 27708, USA. [3] Institute of Fundamental and Frontier Sciences and State Key Laboratory of Electronic Thin Films and Integrated Devices, University of Electronic Science and Technology of China, Chengdu 610054, China. [4] Department of Physics and Astronomy, Ohio University, Athens, OH 45701, USA. [5] Materials Science Division, Argonne National Laboratory, Argonne, IL 60439, USA. [6] Department of Physics, Duke University, Durham, NC 27708, USA. Correspondence and requests for materials should be addressed to M.H.M. (email: m.mikkelsen@duke.edu) or to A.O.G. (email: govorov@ohio.edu) or to G.P.W. (email: wiederrecht@anl.gov)

There is wide-ranging motivation to understand hot electrons in optically excited plasmonic systems for a number of diverse applications. Hot electrons have been demonstrated to inject over large interfacial energy barriers, enabling sensitization of plasmonic Schottky photodetectors to sub-bandgap photons[1–3], nanoscopy with high spatial and chemical sensitivity[4, 5], and photocatalyzed reactions, including hydrogen dissociation on plasmonic nanoparticles[6–9]. The ultrafast optical response from hot electrons has further use in nonlinear optics, ultrafast optical switching, and beam steering[10]. However, what are usually termed "hot" electrons fall into two disparate populations of excited carriers: "nonthermal" electrons with an initial stepwise distribution extending up to the photon energy from the Fermi level and possessing an indefinable temperature, and "thermal" electrons with a quasi-equilibrated Fermi–Dirac distribution and an electronic temperature elevated above that of the surrounding lattice. Due to their relatively low energies, thermal electrons are ineffective for carrier injection and slow to relax. For this reason, nonthermal carriers are of greater relevance for light-harvesting and ultrafast optics applications[3].

Nonthermal electrons form by plasmon dephasing from its wavelike state to a high-energy charge pair in the metal's conduction bands. This process occurs through indirect intraband transitions involving large changes in carrier momentum (Fig. 1a), distinct from the momentum-conserving interband (IB) transitions at energies above the optical bandgap[11]. Due to the breaking of momentum-matching conditions, nonthermal electron generation is predominantly a quantum process driven by optical electric field hot spots and surface-assisted scattering (Supplementary Note 1)[12–15]. Energy is then redistributed among the carriers through electron–electron (e–e) scattering (or Coulomb scattering)[16] to form a thermal electron population (Fig. 1b), which subsequently relaxes through electron–phonon (e–ph) scattering until thermal equilibrium is reached with the surrounding lattice. We note that the diagrams in Fig. 1b represent an ideal case, where an instantaneous pulse excites only nonthermal carriers. However, in reality, both thermal and nonthermal carriers are initially produced during a finite pump pulse (Supplementary Fig. 1), with the relative amount of nonthermal generation determined by the local electric field and the Drude response of the metal.

Nonthermal electrons present considerable challenges to examine and utilize because they are short lived, generally created with low probability, yield only weak perturbations to the overall permittivity of the metal, and exhibit a broad distribution of scattering rates[17]. The broad distribution of scattering rates is a function of their energy; as nonthermal electrons scatter closer to the Fermi level their relaxation rate slows due to Pauli exclusion effects[17]. This is generally described in the context of Fermi liquid theory (FLT), which predicts an e–e scattering rate proportional to $(E-E_F)^2$. This results in values of ~0.1 fs$^{-1}$ at 1 eV energies in noble metals such as Ag (Supplementary Note 2) and can be 2–3 orders of magnitude slower near the Fermi energy $(E_F)$[11, 13, 17–19]. In FLT, the nonthermal e–e scattering is assumed to occur isotropically in a parabolic band structure (Fig. 1a)[17].

In this work, we specifically address the issue of enhancing nonthermal electron generation with the aid of a plasmonic nanopatch metasurface geometry that creates strong electromagnetic hot spots. The resulting high concentrations of nonthermal electrons allow us to study their dynamics in great detail. Such a configuration was previously demonstrated in angle-insensitive perfect absorbers[20, 21], ultrafast emission sources[22–24], single photon emitters[25], and chemical sensors[26, 27]. The use of colloidal silver nanocubes is a simple means of fabricating a conformal metasurface over large areas without the need for nanopatterning. The random orientation of the nanocubes

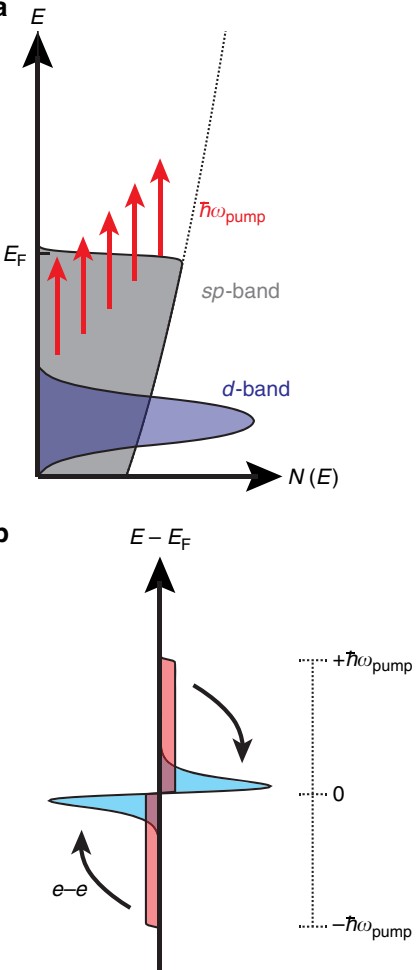

**Fig. 1** Hot electron distributions from intraband excitation. **a** Representative electronic density of states (N) diagram for noble metals under intraband pumping conditions. Free electrons in the sp-band capture the energy of the pump photons ($\hbar\omega_{pump}$) during plasmon dephasing and promote a distribution of electron–hole pairs. **b** Nonthermal (red) carriers initially form a symmetric step-wise distribution relative to the Fermi energy ($E_F$) upon excitation. After electron–electron (e–e) scattering, a thermalized electron population (cyan) with a Fermi–Dirac distribution is formed from the nonthermal electrons with an electronic temperature above that of the lattice

provides an additional advantage in that the optical response is independent of the polarization of incident light[20]. Here we fabricate and optically characterize the steady-state and ultrafast transient response of metasurfaces and experimentally determine the kinetic and spectral response of nonthermal carriers, even when their lifetime is well under 100 fs. Importantly, we propose that the nonthermal carrier decay is anisotropic within the band structure, evidenced by three distinct signatures of the nonthermal response, which exhibit different scattering rates near the X and L points of the Brillouin zone. Theory and modeling support these conclusions.

## Results

**Optical characterization of the metasurface.** The metasurfaces employ silver nanocubes with 150 nm edge lengths that are separated from an underlying gold film by a thin

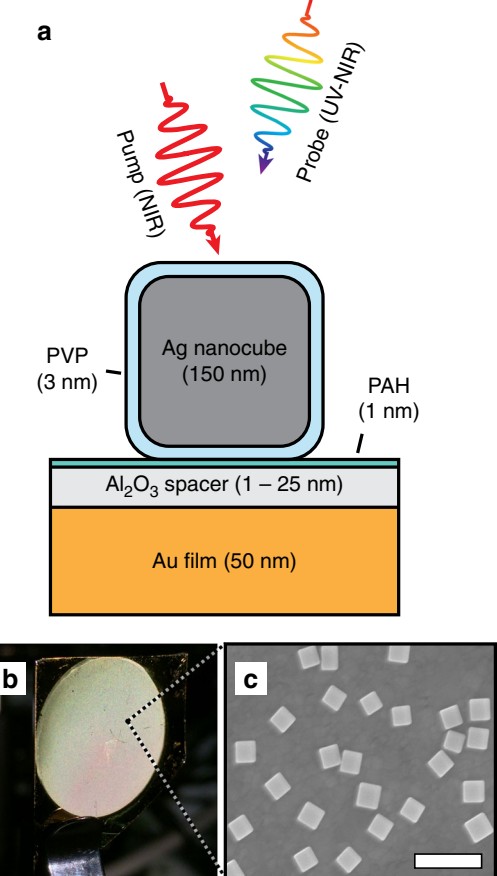

**Fig. 2** Metasurface geometry and characterization. **a** Nanopatch metasurfaces were fabricated by depositing 150 nm (edge length) PVP-coated colloidal silver nanocubes on a 50 nm thick gold film supporting a thin $Al_2O_3$ spacer and interrogated with transient absorption spectroscopy. **b** Image of a metasurface film with an 18 mm diameter and (**c**) corresponding scanning electron micrograph of discrete nanocubes on the surface (scale bar = 500 nm)

polyvinylpyrrolidone (PVP) shell, a poly(allylamine hydrochloride) (PAH) adhesion layer, and an atomic layer deposition (ALD) grown $Al_2O_3$ spacer layer, creating an ensemble of nanopatch antennas (Fig. 2a–c and Supplementary Fig. 2). Samples were fabricated using a range of $Al_2O_3$ spacer thicknesses from 1 to 25 nm. A total of four absorption features are supported in the metasurfaces. These include a substrate-coupled gap plasmon mode in the near-infrared (NIR), a substrate-coupled quadrupolar plasmon mode at ~600 nm, a multipolar plasmon mode at ~400 nm in the Ag nanocubes, and the onset of IB absorption in the gold film at ~500 nm[28]. These absorption features result in spatial distributions of photoinduced surface charge that are modeled by classical 3D electrodynamics calculations (see "Methods" section) and summarized in Fig. 3a. Further simulated electromagnetic field plots are provided in Supplementary Fig. 3. Examples of the steady-state reflectivity measurements and calculated absorbance spectra are shown in Fig. 3b, c for gap thicknesses of 8 and 25 nm. The calculated spectra also show the relative amount of absorption contributions from the Ag and Au components of the metasurface. Additionally, all experimental and calculated absorbance spectra for 1–25 nm gap thicknesses are shown in Supplementary Figs. 4 and 5. The four absorption features combine to provide resonant coverage over a broad spectral range from the ultraviolet (UV) to NIR regions of the spectrum. We find excellent agreement

between experiments and simulations (Fig. 3b, c), and show an inverse dependence of the quadrupolar and gap plasmon resonance wavelengths with the spacer thickness as a result of coupling to the underlying Au film as shown in Supplementary Fig. 4[21, 26].

As can be seen in the steady-state spectra, the full-width-half-maximum of the metasurface gap plasmon resonance is ~200 nm (Fig. 3b, c). This is roughly double the 90 ± 17 nm full-width-half-maximum measured through brightfield absorbance microscopy (see "Methods" section) of individual particles on the metasurface (examples shown in Fig. 3b, inset), indicating some inhomogeneous broadening is present in the ensemble. The simulated absorbance data is also shown in the inset, showing excellent agreement of the calculated linewidth with the experimental data. We attribute this broadening to a finite distribution in particle sizes (Supplementary Fig. 2) that extends the coverage of the gap resonance in the NIR.

**Hot electron production.** Many time-resolved experiments have been performed on various nanoparticle geometries[17, 19, 29–31], however in each case, their transient response was dominated by the decay of thermal electrons on the picosecond timescale. Only in a recent gap mode structure has the ultrafast growth and decay of nonthermal carriers been significant[12]. To confirm the improved nonthermal carrier generation in the nanopatch metasurface geometry, we first experimentally compare transient absorption kinetics at the gap mode in samples with an 8 nm spacer to a thicker 25 nm spacer and a film of bare Ag nanocubes on an $SiO_2$ substrate, as shown in Fig. 4a, b. (A detailed discussion of our transient absorption measurements will be addressed in the following section). As can be seen, the ultrafast (<300 fs) decay attributed to nonthermal carriers is strongest for the 8 nm spacer, reduced for the 25 nm spacer, and is nonexistent for nanocubes on $SiO_2$ without the metasurface geometry.

The observation of the unusually strong ultrafast response in the kinetics in our samples can be explained qualitatively by calculating the rates of nonthermal carrier generation. A simple quantum formalism describing the rate of optical generation of nonthermal carriers involves integration over the surface area of a nanoparticle:

$$\text{Rate}_{\text{nonthermal}} = \sum_{i=\text{Ag,Au}} \frac{2e^2 E_{\text{F},i}^2}{\pi^2 \hbar (\hbar\omega)^3} \int_{S_{\text{NC},i}} |E_n(\theta,\varphi)|^2 \, dS, \quad (1)$$

where $E_n$ is the normal electric field inside a nanocrystal near the surface, $E_F$ is the Fermi energy of the metal, and the integral in Eq. 1 is taken over the surface of a nanocrystal ($S_{\text{NC}}$)[32]. Such calculations (Fig. 4c) clearly indicate that the generation of hot carriers becomes strongly amplified at the plasmonic wavelength in samples with small gaps due to the formation of plasmonic hot spots. Moreover, we can estimate an averaged number of energetic electrons (i.e., electrons in the interval $E_F < E < E_F + \hbar\omega$) during a short excitation pulse (80 fs is assumed here to match the experiment) as:

$$N_{\text{Nonthermal,avg}} \approx \sum_{i=\text{Ag,Au}} \text{Rate}_{\text{Nonthermal},i} \cdot \overline{\tau}_{e-e,i} \left( 1 - \frac{1 - e^{-\Delta t/\overline{\tau}_{e-e,i}}}{\Delta t/\overline{\tau}_{e-e,i}} \right),$$

$$(2)$$

where $\overline{\tau}_{e-e,i}$ is the characteristic e–e scattering lifetime of an energetic electron in the corresponding metal ($i$ = Ag, Au) and $\Delta t$ is the pulse duration. The Coulomb scattering time used above is defined as $\overline{\tau}_{e-e,i} = 4\tau_{0,i} \cdot E_{\text{F},i}^2 / (\hbar\omega)^2$, where the material constant $\tau_{0,i}$ is given by Supplementary Eq. (5). The above equations were written for a steady-state regime under CW illumination since the

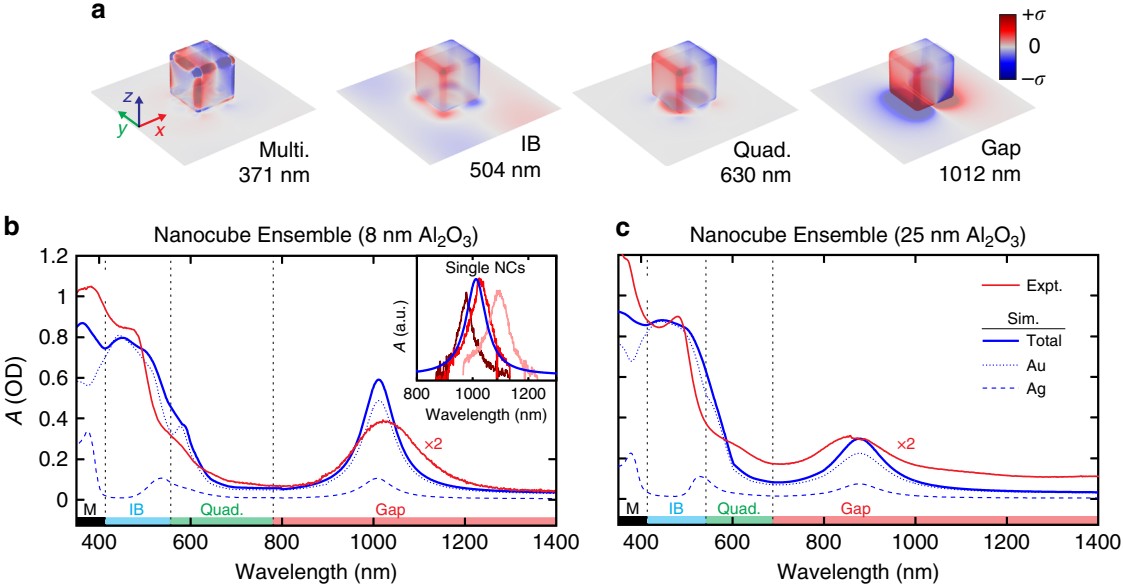

**Fig. 3** 3-D maps of surface charge distributions with experimental and calculated absorbance spectra. **a** Surface charge distributions (arbitrary units) for the 8 nm $Al_2O_3$ sample at the indicated wavelengths corresponding to the multipolar mode (M), the gold interband transition (IB), the quadrupolar mode (Quad), and the gap plasmon mode (Gap) calculated with COMSOL assuming normally incident light polarized along the x-direction. Steady-state absorbance is simulated for samples with an 8 nm (**b**) or 25 nm (**c**) $Al_2O_3$ spacer assuming a nanocube density of $4\,\mu m^{-2}$ and compared to the experimental ensemble response of the metasurface. The calculated absorbance is also broken down into the relative contributions to absorbance from the Ag and Au components. The inset to **b** is a comparison of three experimental absorbance spectra of single nanocubes (NCs) at the gap mode to the simulated spectrum, showing an excellent agreement between homogeneous linewidths

optical pulse duration is longer than the plasmonic relaxation time. For calculations, we use a flux of $2.5 \times 10^8\,W\,cm^{-2}$ that is typical for our experiments. As can be seen in Fig. 4d, it is the quantity $N_{\text{Nonthermal,avg}}$ that directly creates the ultrafast non-linear response, which is highly enhanced by the gap plasmons in the NIR range. In other words, the NIR hot spots govern the population of energetic nonthermal electrons with energies in the interval $E_F < E < E_F + \hbar\omega$. The key parameter here is the plasmonic energy that is reduced for the red-shifted gap plasmon modes. There are two physical mechanisms responsible for the favorable features of hot electrons for the gap plasmon: (1) the quantum amplitude $(\hbar\omega)^{-3}$ in Eq. 1, and (2) electrons in the NIR gap plasmons have lower energies and therefore are much longer lived since $\overline{\tau}_{e-e} \propto (\hbar\omega)^{-2}$ (Eq. 2 and supplementary Eq. (1b)). When energetic electrons become longer lived, the population (Eq. 2) strongly increases. We also note that the gap plasmon also has increased absorption cross sections (theory data in Supplementary Note 1 and Supplementary Fig. 5). Basically, we can see that certain aspects of the gap plasmon optical response are amplified in the NIR range.

To further highlight the benefits of the nanopatch metasurface geometry, we benchmark its nonthermal electron generation rate against other nanoparticle configurations through electrodynamic simulations (Fig. 5 and Supplementary Fig. 6). For these calculations, we evaluate the peak generation rate when excited at the gap plasmon resonance (and other resonances when multiple modes are supported in the structure). In all cases, the nanoparticle volume was fixed at $(150\,\text{nm})^3$ to remain consistent with our nanocube metasurface samples (Fig. 2a and Supplementary Fig. 2). We find up to ~10× higher production in the metasurface geometry (configurations 5 and 7–10) compared to bare nanospheres, nanorods, or nanocubes (configurations 1–4 and 6). The highest rates of production in the metasurface geometry arise from the electromagnetic hot spot generated

within the film-nanocube gap. We also observe a ~20% higher nonthermal electron generation rate when employing Ag instead of Au nanocubes in the same geometry (configurations 9 and 10). This can be attributed to less damping and a longer momentum relaxation time in the Ag nanocubes[32]. Additionally, metasurfaces with thinner spacer layers exhibit higher nonthermal electron generation rates owing in part to an increased optical field strength (Supplementary Note 1 and Supplementary Fig. 7). These results are consistent with the previous literature[13, 14], where nonthermal carrier generation was shown to be a surface effect enhanced by strong optical fields and hot spots. For example, quantum efficiencies as high as 90% have been predicted for nonthermal electron generation in dimer nanostructures with small gaps[33].

**Transient absorption spectroscopy.** Femtosecond transient absorption measurements were performed to measure the ultra-fast response of the plasmonic metasurface (Fig. 6a). In all cases, samples were excited by the NIR pump at the gap plasmon resonance and probed with a continuum pulse spanning the UV-NIR (Fig. 2a). For clarity, we begin by focusing our discussion on samples with an intermediate (8 nm) thickness and later extend our analysis to other spacer thicknesses. For the 8 nm sample, the 1100 nm pump wavelength (1.13 eV) only excites intraband transitions in the conduction bands of both the Au film and Ag nanocubes, owing to their much higher IB transition energies of ~2.5 and ~4.0 eV, respectively. We observe four differential absorption features well matched to the steady-state absorption features (blue and purple curves in Fig. 6b, respectively). Figure 6c, d shows cross-sections of the field profiles and induced surface charges at each corresponding peak. In the time domain, the sample exhibits kinetics ranging from ultrafast pulsewidth-limited decays with time constants <300 fs to long-lived (~ns) coherent acoustic phonon modes (Supplementary Note 7).

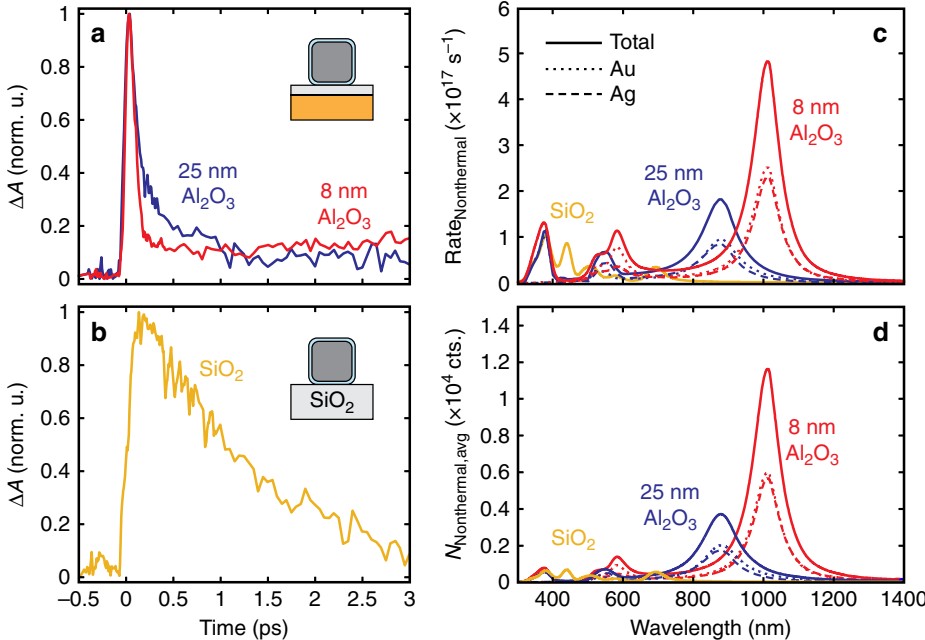

**Fig. 4** Enhanced generation and detection of nonthermal electrons in metasurface geometry. **a** Kinetic trace of normalized differential absorbance for Ag nanocubes on an 8 nm $Al_2O_3$ spacer compared to a 25 nm $Al_2O_3$ spacer at the gap resonance and (**b**) to bare Ag nanocubes on $SiO_2$ in transmission mode, which exhibits no ultrafast (~100 fs) decay of the response. The 8, 25 nm, and $SiO_2$ samples were, respectively, pumped/probed at 1100/1120, 900/920, and 500/365 nm with incident fluences of 40, 40, and 500 μJ cm$^{-2}$ (absorbed fluence kept constant). The ~100 fs response arises from the relaxation of highly energetic nonthermal carriers. **c**, **d** Comparison between nonthermal carrier generation rate (**c**) to estimates of the peak nonthermal carrier density (**d**) as a function of geometry and pump wavelength. A much larger contribution to the signal is expected for excitations in the NIR

Representative kinetic traces probing at the four absorption features are shown in Fig. 6e. From them, it is apparent that the strongest ultrafast (<300 fs) response occurs out in the red to NIR wavelengths resonant with the quadrupolar and gap plasmon modes. At these wavelengths, intraband transitions are efficiently excited by both the pump and the weak probe (Fig. 1a). Consequently we expect the signal to be dominated by nonthermal carriers and decay during e–e scattering. In contrast, higher energy excitations near the multipolar mode of the Ag nanocubes and the gold IB transition are expected to be less sensitive to nonthermal carriers. In particular, the relatively small contribution of the ultrafast component in the gold IB response stems from two factors. First, the shorter probe wavelengths sample the bulk response from the entire Au film (Fig. 3a). In contrast, the gap, quadrupolar, and multipolar plasmon resonances are locally confined near the nanocubes, where the nonthermal electron density is initially concentrated (Fig. 6c, d). Second, the IB response selectively probes electronic transitions from the d-band to the edge of the Fermi level, and is therefore less sensitive to high energy nonthermal electrons.

**Separation of scattering processes**. To separate out the kinetics and spectral signatures of the individual scattering processes (e–e, e–ph, and phonon–phonon (ph–ph)) from our transient absorption data, we performed a lifetime density analysis (LDA) of the 2D maps (Fig. 7). The LDA method, while computationally intensive, provides a model-independent determination of the constituent lifetimes present in the temporal response along with their amplitudes (Supplementary Note 3)[34]. Furthermore, it allows the separation of pulsewidth-limited responses from the data when the instrument response is well characterized, as is the case here (Supplementary Fig. 8).

Figure 7a shows the corresponding lifetime density map (LDM) obtained from the transient absorption data in Fig. 6a

using LDA. Regardless of probe wavelength, oscillations are present in the amplitude of the LDM, indicating peaks in the lifetime distribution. Vertical dashed lines designate the cutoff wavelengths between absorption features (plasmon modes or IB transition), where the signal reaches a minimum and exhibits isosbestic points. The wavelength range for each was analyzed independently to extract the corresponding spectral (Fig. 7b, c) and kinetic (Figs. 7d–g) components. As is generally reported[17–19], we observe a peak with a ~1 ps lifetime corresponding to thermal e–ph scattering, a peak with a ~10 ps lifetime corresponding to ph–ph scattering into coherent acoustic modes, and a long decay of ~100 ps corresponding to a spectral shift due to lattice heating.

Surprisingly, we also observe three distinct ultrafast kinetic components corresponding to nonthermal e–e scattering, which we label in Fig. 7d–g according to their respective rates (fast, intermediate, and slow). Across the three plasmon resonances, we observe consistent peaks in each lifetime distribution at $25 \pm 5$, $105 \pm 20$, and $246 \pm 60$ fs, which we find to be independent of pump fluence (Supplementary Fig. 10a–c). This further confirms their nonthermal nature, as the lifetime of thermal electrons is fluence-dependent and proportional to the electron temperature rise relative to the lattice (Supplementary Note 2)[17, 19]. Additionally, we find the "fast" component cannot be attributed to the initial plasmon dephasing rate. Since the nanopatch antennas primarily exhibit nonradiative damping of the gap plasmons[35], the single nanocube homogeneous linewidths of $103 \pm 21$ meV are a direct measure of the dephasing rate (Fig. 3b, inset). This translates to a $13 \pm 2.4$ fs dephasing time, consistent with the known intraband nonradiative damping of gold nanorods, which is half the lifetime of the fast carriers[36].

Although the metasurface comprises both Au and Ag, several factors indicate the kinetics of the three plasmon resonances correspond to carriers predominantly residing in the Ag

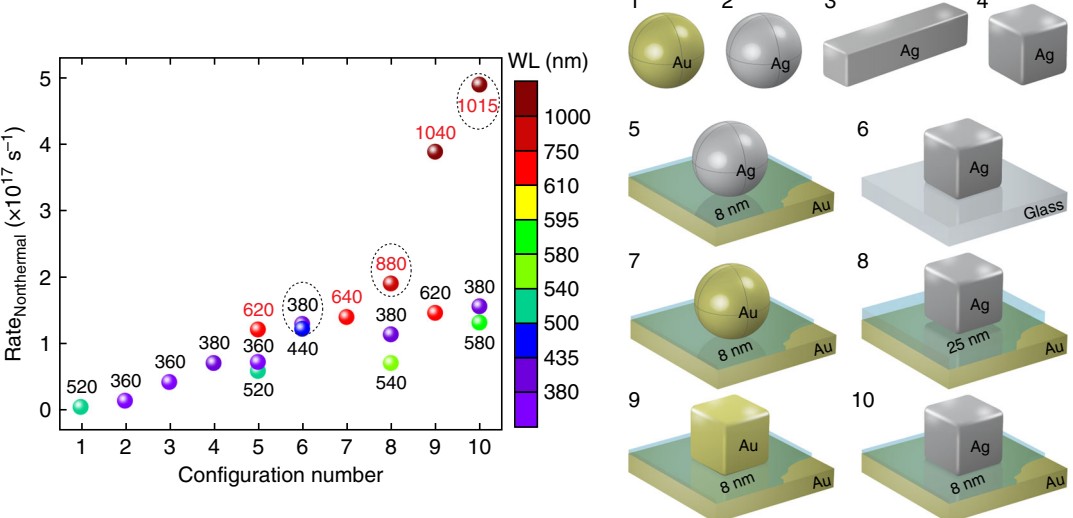

**Fig. 5** Geometry dependence of nonthermal hot electron generation. Nonthermal carrier generation rates for the ten displayed nanoparticle configurations are plotted with their corresponding plasmon resonance wavelengths, which are labeled and color-coded for clarity. Red text corresponds to gap plasmon resonances. Geometries employing Ag exhibit higher rates than Au owing to lower damping and a longer momentum relaxation time, and generation through gap excitation is shown to be much more efficient than bare nanoparticle resonances. In all cases the nanoparticle volume was fixed at $(150 \text{ nm})^3$, corresponding to a diameter of 185 nm for the nanospheres and a 340 nm length (100 nm edge) for the nanorod. In the nanopatch geometries (5, 7–10) an $Al_2O_3$ spacer was employed with the corresponding thickness labeled

nanocubes. First, the optical fields at the multipolar mode are uncoupled from the gold film and localized on the nanocubes (Figs. 3a and 6c, d; and Supplementary Fig. 3). This ensures any spectral shifts of the multipolar mode are predominantly sensitive to the Ag permittivity. Second, the slowest e−e scattering rate closely matches the previously estimated 220 fs thermalization time for Ag films in the weak perturbation regime[17]. Third, while the Ag nonthermal electrons are confined within the nanocube volume, those in Au are free to diffuse upon generation. The ballistic transport velocity of optically-excited nonthermal carriers in Au has been previously measured to be at least $1 \text{ nm fs}^{-1}$, indicating they should rapidly deplete from the gap region of the nanopatch antennas and reduce their local concentration after excitation[37].

Figure 7b, c further shows the decay-associated spectra (DAS) for the multipolar and gap plasmon resonances. The DAS describe the change in the transient absorption signal for each scattering process. Importantly, we observe distinct DAS for all three nonthermal carrier subpopulations, as well as for the thermal carriers (e−ph). Just like the kinetics, we find the DAS e−e lineshapes are fluence-independent (Supplementary Fig. 10d–i), strongly supporting their assignment to nonthermal carriers. However, the magnitude of the contribution to the overall signal from each subpopulation varies with pump power (Supplementary Fig. 11). With increasing fluence, the fast and intermediate nonthermal carriers have higher relative contributions to the signal than the slower nonthermal carriers. This yields an apparent acceleration of the overall e−e scattering rate of nonthermal carriers (thermalization time) and an inversion of the differential absorbance spectrum at early times in the gap mode between low ($20 \text{ μJ cm}^{-2}$) and high fluence ($130 \text{ μJ cm}^{-2}$). Previous studies have characterized this as a transition from a "weak to strong-perturbation regime"[17]; however, until now the origin of this behavior has remained elusive.

**Signatures of anisotropic behavior**. We propose that the three e−e decay components can be assigned to nonthermal carriers

localized near the X, L, and K symmetry points in the band structure, as illustrated schematically in Fig. 8a, b. The concept of nonthermal carriers being distinct in different portions of the band structure, i.e., anisotropic, represents a departure from the bulk treatment of noble metals as isotropic and has recently begun to gain traction in theoretical work (Supplementary Note 2)[13, 38]. First-principles calculations of nonthermal carrier generation and scattering have predicted a range of scattering rates within the Brillouin zone arising from variations in the band curvature (effective electron mass or intraband plasma frequency)[13, 38], which is consistent with the ~10× variation in e−e scattering rates measured here. In the case of plasmonic intraband excitation, the breaking of momentum matching constraints enables the excitation of nonthermal carriers with large wavevectors along any band crossing of the Fermi surface. For noble metals such as Ag and Au, this implies a joint excitation of carriers at the X, K, and L symmetry points.

Until now, we have focused our analysis on the three plasmon resonances of the metasurface, which indiscriminately couple to intraband transitions in the Ag nanocubes (blue arrows in Fig. 8b). As a control experiment in support of our observations for Ag nanocubes, we now consider the response of the gold IB transition, where the e−e and e−ph scattering DAS provide additional support for anisotropic interpretation of the data (Fig. 8c). Unlike the purely intraband transitions probed by the plasmon modes (blue arrows in Fig. 8b), the gold IB response selectively probes carrier dynamics at specific points of the band structure (blue arrows in Fig. 8a). In our measurement window, IB transitions near the Fermi edge of the X and L-points are excited at peak energies of 2.4 eV (517 nm) and 2.8 eV (443 nm), respectively[39]. For thermal carriers (e−ph) and slow nonthermal carriers (e−e$_{slow}$), we observe a strong transient bleach or absorption feature, respectively, at both 436 and 517 nm, closely matching the literature values for IB transitions at the X and L-points. However, the intermediate nonthermal carriers in Au only exhibit a bleach feature near 517 nm, indicating their response is localized near the X-point. Finally, the fast nonthermal carriers exhibit bleach components near the X and L-points, however, the

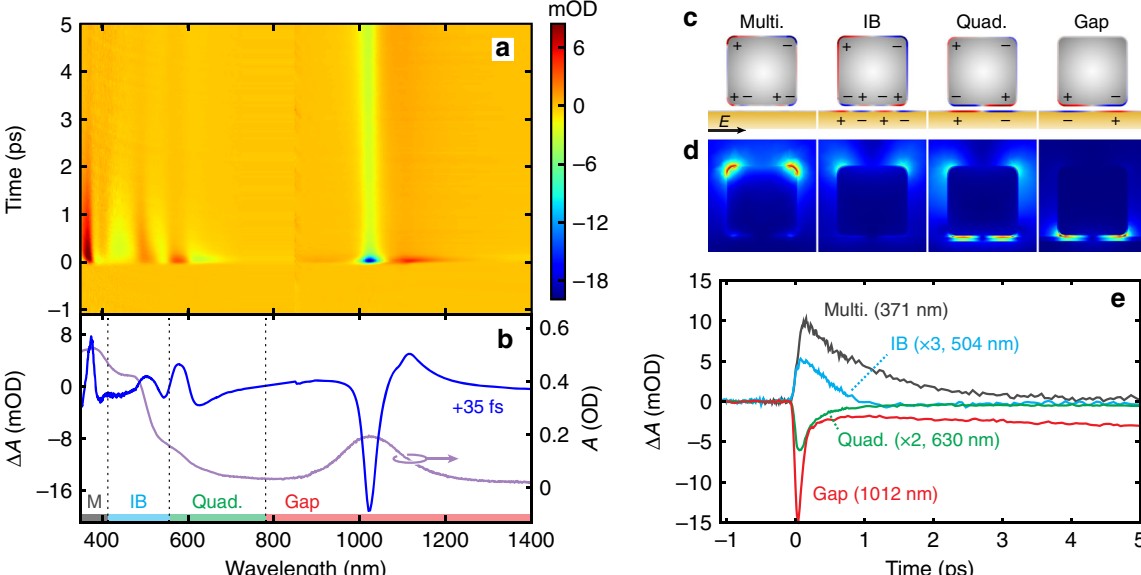

**Fig. 6** Transient absorption measurements of the metasurface spanning the UV to NIR. **a** Differential absorbance spectral map of the 8 nm $Al_2O_3$ metasurface pumped at 1100 nm with a fluence of ~130 μJ cm$^{-2}$. **b** Steady-state absorption spectrum (purple) and differential absorption spectrum (blue) taken at +35 fs relative to the pump pulse with features corresponding to the nanocube multipolar, quadrupolar, and gap plasmon resonances and the gold IB transition as indicated. The vertical dotted lines correspond to the zero-crossing points between each transition. Cross section of the surface charge (**c**) and electric field profiles (**d**) at the peaks of the IB transition and plasmon modes for an electric field oriented in-plane. The field strength scale is up to 20× free space for the multipolar (Multi) mode, interband (IB) transition, and quadrupolar (Quad) mode and 100× free space for the gap mode. **e** Corresponding kinetic traces for the four absorption features indicated in (**b**) at select wavelengths. The multipolar and IB kinetics show a response dominated by thermal electron–phonon scattering, while the quadrupolar and gap plasmon resonances exhibit much faster responses arising from nonthermal e–e scattering

response near the X-point is shifted to lower energy and is indicative of a carrier population below the Fermi level. We expect that anisotropic carrier scattering also occurs near the K-point, but this transition lies beyond our measurement window.

Compared to the plasmon resonances in the silver nanocubes, the gold IB transitions exhibit much slower rates of e–e scattering overall (Fig. 7f). We observe fluence-independent peaks at lifetimes of $46 \pm 13$, $172 \pm 4$, and $338 \pm 71$ fs. The reduced scattering rate arises from a stronger charge screening by bound $d$-band electrons in the gold and is roughly proportional to $1/\sqrt{\varepsilon_\infty}$ at lower carrier energies, where $\varepsilon_\infty$ is the static permittivity of the metal. We find the slow nonthermal e–e scattering time to be ~37% longer in gold, consistent with the predicted value of 35%[17]. Thus, the combination of observing 3 distinct population decays in Au that are slower than Ag by the predicted value, combined with the additional spectral evidence made available only in Au, is a strong support for anisotropic nonthermal hot electron decay in these metasurfaces.

We now extend our analysis to samples with a range of spacer thicknesses (Fig. 9) both pumped and probed at the gap plasmon resonance. Since the gap resonance wavelength is inversely dependent on spacer thickness (Supplementary Fig. 4), the pump energy ($\hbar\omega_{pump}$) is varied to excite each sample on-resonance. Figure 9 shows the dependence of the peaks of the nonthermal e–e lifetime distributions as a function of $(\hbar\omega_{pump})^{-2}$ for each spacer. We observe an increase in the peak e–e lifetimes with $(\hbar\omega_{pump})^{-2}$ for both the fast and slow subpopulations. As the average initial nonthermal carrier energy scales with $\hbar\omega_{pump}$, this trend is consistent with FLT. However, we observe an opposite trend for intermediate carriers. Since the e–ph coupling rate is expected to be constant in these energy ranges[13], we ascribe this to nonthermal e–e scattering in a region with increasing effective

mass (reduced plasma frequency) at higher energies (Supplementary Note 2). To confirm that these trends are solely pump energy dependent and not arising from the antenna geometry, we also excited the 5 nm $Al_2O_3$ sample at multiple energies about the gap resonance and observe the same result. The different trends in lifetime as a function of energy support our conclusion that anisotropic electron scattering creates subpopulations at different points in the band structure.

## Discussion

A combination of factors likely masked the detection of these nonthermal subpopulations in earlier literature. In uncoupled nanoparticles and metal films, for example, a lack of field hot spots significantly reduces nonthermal carrier generation rates as compared to the nanopatch metasurface. Additionally the analysis of kinetics at single wavelengths (particularly near the IB transitions) biases the response, and at many wavelengths contributions from two of the nonthermal carrier populations can cancel out (Supplementary Note 6 and Supplementary Fig. 11). This can incorrectly give the impression of wavelength-dependent e–e scattering rates when, in fact, a global analysis reveals that different wavelengths measure different percentages of contributions from nonthermal carriers. Finally, the extraction of rate distributions from complex data sets requires the kinds of global analysis methods such as the LDA employed in this work.

In summary, we have measured the optical response of both nonthermal and thermal hot electrons in a plasmonic metasurface using ultrafast transient absorption spectroscopy. This was enabled by the high generation efficiency of nonthermal carriers in the nanopatch geometry, which we showed is a result of the hot spot generated within the gap. Our results show that the choice of geometry and materials are crucial for producing strong ultrafast

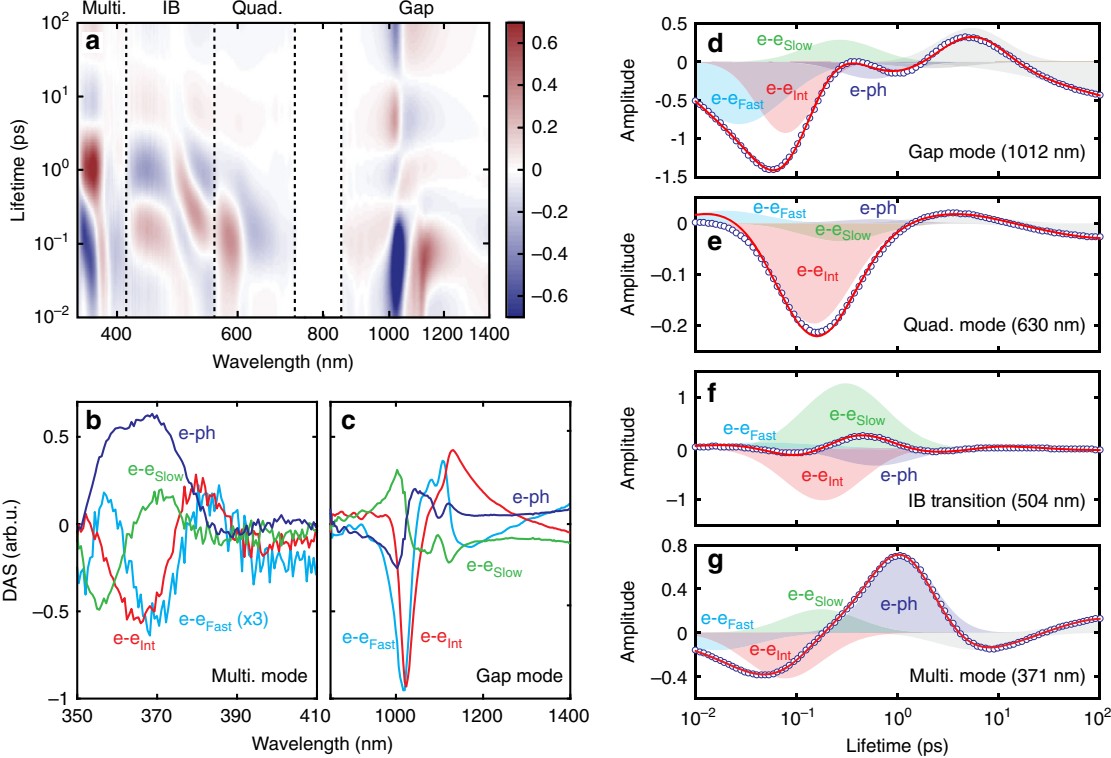

**Fig. 7** Lifetime density analysis of the ultrafast response. **a** Lifetime density map (LDM) fitted to the differential absorbance data in Fig. 6 and displayed on a log–log scale. Multiple distinct peaks can be observed spanning the entire range of lifetimes at each resonance. **b**, **c** Comparison of the decay-associated spectra (DAS) corresponding to fast, intermediate (int), and slow nonthermal e–e scattering and thermal e–ph scattering at the multipolar and gap plasmon resonances. The small fluctuations at 1100 nm are artifacts from residual pump scatter. **d**–**g** Lifetime traces of the LDM (blue circles) at the same wavelengths as the kinetics in Fig. 6e. Contributions from nonthermal electron–electron scattering and thermal electron–phonon scattering are indicated (shaded regions) along with the total fitted response (red line). The gray regions are contributions from phonon–phonon scattering (~10 ps) and semi-infinite decay due to lattice heating (~100 ps)

nonthermal electron contributions that can be measured through transient absorption. We present experimental evidence of three subpopulations of nonthermal carriers with distinct e–e scattering rates and spectra. We propose that these subpopulations are localized near band crossings of the Fermi surface and decay anisotropically with different time constants. This work supports recent first-principles calculations, predicting variations in non-thermal carrier scattering near the Fermi surface of noble metals[13, 38]. The nonthermal carrier response is shown to extend from the UV to NIR, spanning multiple surface plasmon resonances of the nanopatch antennas and IB transitions of the underlying gold film. We find an order of magnitude variation in e–e peak scattering times of the three nonthermal subpopulations spanning ~25–250 fs in the Ag nanocubes and ~45–340 fs in the Au film. Harnessing this ultrafast carrier response, in particular at intraband transitions, could enable low-power optical switching well into THz frequencies. Furthermore, one may selectively excite transitions in regions of the band structure with longer lifetimes to leverage the wavevector-dependence of nonthermal carrier scattering, which could improve the efficiency of hot carrier injection. The new insights provided here can impact a diverse range of fields, including telecommunications, nonlinear optics, metamaterials, photocatalysis, photodetectors, and the study of systems far from thermal equilibrium.

## Methods

**Sample fabrication**. Silicon substrates with a 50 nm gold film were coated with ALD-deposited alumina ($Al_2O_3$) spacers of varying thickness (1–25 nm) by VaporPulse Technologies. PVP-coated silver nanocubes with ~150 nm edge length

(nanoComposix, Inc.) were deposited by a previously reported method[40]. Briefly, a single PAH layer was deposited on the $Al_2O_3$ surface to promote nanocube adhesion. The PAH with an average molecular weight of 58,000 was purchased from Sigma-Aldrich. A solution of 150 nm nanocubes was then prepared by concentrating a 1 mg mL$^{-1}$ solution of nanocubes in ethanol and re-suspending in 18 MΩ DI water, yielding a final concentration of 4 mg mL$^{-1}$. An 18 mm round coverslip was placed on top of 15 μL of solution, and the nanocubes were allowed to settle onto the surface for 30 min. The coverslips and excess solution were then removed with DI water and the samples were dried with nitrogen gas.

**Nanocube characterization**. Samples were imaged under high vacuum (<1 × 10$^{-4}$ Torr) with a JEOL 7500 scanning electron microscope. To measure nanocube dimensions, the area, and min and max Feret's diameters were extracted using the particle analysis macro in ImageJ. From these values, the edge lengths and radius of curvature for each particle were fit assuming a rounded rectangle shape (Supplementary Fig. 2). Over 440 individual particles were analyzed to obtain reliable statistics.

**Reflectivity measurements**. Absolute ground-state reflectivities were measured using the integrating sphere of a Perkin Elmer Lambda 950 spectrometer coupled to a PbS photodetector and PMT (Fig. 3b, c and Supplementary Fig. 4). Single nanocube absorption spectra (Fig. 3b inset) were captured using an inverted Olympus IX71 microscope coupled to a Princeton Instruments spectrometer and InGaAs detector (Acton SP2300 and NIRvana 640, respectively). Unpolarized light from a halogen lamp was both focused and collected through a 100× objective (Olympus LMPlan IR, NA = 0.8) and passed through a 75 μm slit prior to entering the spectrometer to isolate individual particles. Spectra of the specular reflected light from each nanocube were integrated for 200 ms, averaged over 100 exposures to avoid detector saturation, and were normalized to spectra of the bare substrate between particles. Single particle spectra were then fitted to a Lorentzian function to extract the homogeneous linewidths.

**Transient absorption spectroscopy**. Femtosecond transient absorption spectroscopy was performed using a Ti:sapphire laser with a regenerated 800 nm output

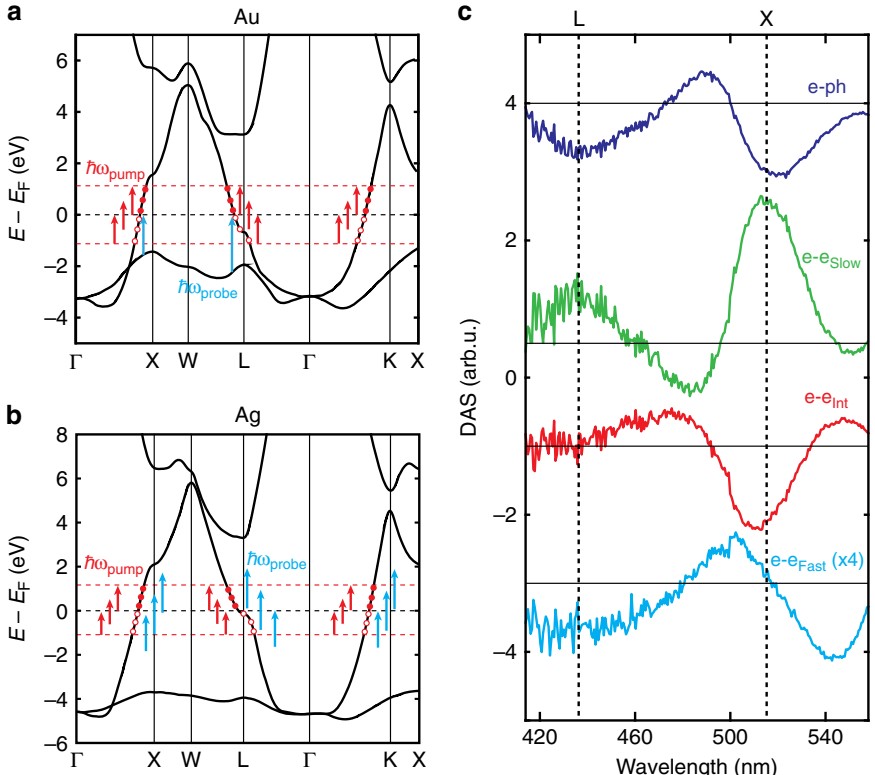

**Fig. 8** Resolving carrier localization within the Au band structure from IB transitions. **a** Band structure of Au exhibiting three *sp*-band crossings of the Fermi level near the X, L, and K symmetry points of the Brillouin zone. The intraband pump (red arrows) at the gap resonance promotes nonthermal carrier distributions (circles) in the *sp*-bands crossing the Fermi surface. The IB probe wavelengths (blue arrows) only monitor transitions from the upper *d*-band to crossings of the Fermi surface near the X and L points. **b** Band structure of Ag exhibiting strong similarities to that of Au. IB transitions in Ag occur at energies higher than ~4 eV, thus the probe (blue arrows) monitors only intraband transitions in the measurement range. The red arrows again refer to the intraband pump photons at the gap plasmon resonance. The data for (**a**, **b**) were taken from literature[42, 43], and only the highest energy *d*-band and first two *sp*-bands are shown for clarity. **c** Decay-associated spectra (DAS) in the IB region of the 8 nm $Al_2O_3$ sample pumped at ~130 µJ cm$^{-2}$. Each DAS is offset for clarity. The fast nonthermal carriers redshift the X transition, indicative of carriers residing below the Fermi level near the X point, and also exhibit a bleach at the L transition. Intermediate nonthermal carriers appear to be localized solely at the X point where they induce a bleach of the absorbance. Slow nonthermal carriers yield perturbations to transitions near both X and L points. IB transitions at the K point occur at much higher energies, and thus are not resolved in the measurement range

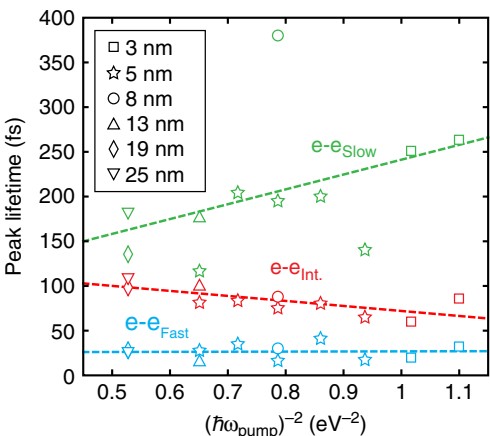

**Fig. 9** Spacer and pump energy dependence of nonthermal response at the gap mode. Shift in the lifetime distribution peaks corresponding to nonthermal e–e scattering as pump energy ($\hbar\omega_{pump}$) and $Al_2O_3$ spacer thickness are varied. An opposite trend is observed for the intermediate carriers. The different dependence of peak lifetime vs. pump energy of the three e–e components supports their assignment to distinct nonthermal electron populations whose decay is not isotropic. The dashed lines are guides to the eye

pulsed at 5 kHz, 95% of which was coupled to an OPA to produce a pump beam with tunable wavelength between 900 and 1300 nm from the frequency-doubled idler. The remaining 5% of the regeneration pulse was sent through a delay line with <10 fs temporal resolution and focused onto a $CaF_2$ (sapphire) crystal to generate a continuum probe pulse extending from 350 to 750 nm (850–1600 nm). The pump pulse was chopped at 2.5 kHz, and both beams were focused at near-normal incidence onto a ~350 µm diameter spot and spatially overlapped on the sample. Both pump and probe were cross-polarized by inserting a half-wave plate into the path of the pump beam prior to the sample. A Glan–Taylor polarizer was then inserted after the sample in the path of the reflected continuum probe to isolate the probe light and minimize pump scatter. The reflected beam was fiber-coupled to a spectrometer with a Si (InGaAs) array detector for UV-Vis (NIR) measurements, and the difference spectrum was recorded as a function of delay time. The cross-correlated pulsewidth of the two beams was determined, respectively, to be $81.1 \pm 5.7$ and $83.2 \pm 5.8$ fs from the optical Kerr effect in diamond at UV-Vis and NIR probe wavelengths (Supplementary Fig. 8), closely matching the $75.1 \pm 11.8$ fs value extracted from LDA fitting.

**Transient absorption analysis.** Scattered light was first subtracted from the background of each scan prior to chirp correction of the data using SurfaceXplorer (Ultrafast Systems). For chirp correction, time zero values were initially taken to be the point at which the signal diverges from the background and were later set to the pump arrival time after fitting. LDA was then performed on the data sets using the Optimus software package, which employs the L-curve method for accurate fitting[34]. The data were split into three wavelength ranges: 350–500, 500–730, and 850–1400 nm to accommodate the different signal-to-noise levels across the probed spectrum. Regularization factors were sampled on a log-scale, and density maps were taken at the corner of the L-curve as is standard practice (Supplementary Fig. 9). The LDM range corresponding to each transition was then globally fit to a

set of normal distributions to extract the lifetime distribution and corresponding DAS for each scattering process. Further details on the LDA and global fitting procedures can be found in Supplementary Notes 3 and 4.

**Classical electromagnetic simulations**. Classical simulations of the nanopatch structures were performed using classical electrodynamics with COMSOL Multiphysics, assuming standard boundary conditions and using literature values for the dielectric functions of the gold film and silver nanocubes[41]. Calculations assumed a normally incident beam and periodic boundary conditions with the lateral period taken as a large number (500 nm) to avoid interparticle coupling. Silver nanocubes with an edge length of 154 nm and a corner radius of 10 nm (taken from measurements) were modeled with a 3 nm PVP coating, and the surface of the $Al_2O_3$ spacer was assumed to be covered with a 2 nm PAH layer. The refractive indices for PVP, PAH, and $Al_2O_3$ were taken to be 1.52, 1.4, and 1.77, respectively. Using these conditions, we achieved excellent agreement with experiment for the positions of the plasmon resonances and the homogeneous linewidths (Fig. 3b, c and Supplementary Fig. 4b).

The local dissipation spectra and maps in the text were computed from the standard equation for the local rate of losses:

$$Q_{abs,local} = \langle \mathbf{j} \cdot \mathbf{E} \rangle_t = \text{Im}(\varepsilon_{metal}) \frac{\omega}{8\pi} \mathbf{E}_\omega \cdot \mathbf{E}_\omega^*, \tag{3}$$

where $\mathbf{E}_\omega$ and $\varepsilon_{metal}$ are the complex field amplitude and dielectric constant of the corresponding metal, respectively. Another parameter illustrating characters of the plasmonic modes in our sample is the surface charge shown in Figs. 3a and 6c. It was calculated in the following manner:

$$4\pi\sigma_\omega = E_{n+} - E_{n-} \tag{4}$$

where $E_{n+}$ and $E_{n-}$ are the normal fields near the surface inside and outside the metal, respectively.

**Data availability**. The data that support the plots within this paper and other findings of this study are available from the corresponding author upon reasonable request.

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

## Acknowledgements

This work was performed, in part, at the Center for Nanoscale Materials, a U.S. Department of Energy Office of Science User Facility, and supported by the U.S. Department of Energy, Office of Science, under Contract No. DE-AC02-06CH11357. J.W.S., G.M.A. and M.H.M. acknowledge support from the Air Force Office of Scientific Research Young Investigator Research Program (AFOSR, Grant. No. FA9550-15-1-0301). A.O.G. was supported by the Army Office of Research (MURI Grant W911NF-12-1-0407) and the Volkswagen

Foundation (Germany), and via the Changjiang Chair Professorship (China). X.-T.K. was supported by the oversea postdoc program of Institute of Fundamental and Frontier Sciences at the University of Electronic Science and Technology of China, and by Changjiang Scholar funding. Work by A.B.f.M. was supported by Argonne-Northwestern Solar Energy Research (ANSER) Center, an Energy Frontier Research Center funded by DOE, Office of Science, BES under Award # DE-SC0001059. We would like to thank Dr. Tal Heilpern and Dr. Rich Schaller for their helpful discussions.

## Author contributions

M.E.S., G.M.A., M.H.M., A.O.G. and G.P.W.: Conceived and planned the work. J.W.S., G.M.A., A.B.F.M., and D.R.: Fabricated the samples. M.E.S., D.J.G. and G.P.W.: Performed the optical spectroscopy. M.E.S., A.O.G. and G.P.W.: Performed the data analysis. X.-T.K., Z.W. and A.O.G.: Performed quantum and classical electrodynamics calculations. M.H.M., A.O.G. and G.P.W.: Supervised the project. M.E.S., M.H.M., A.O.G. and G.P.W.: Wrote the manuscript with contributions from all authors.

## Additional information

**Competing interests:** The authors declare no competing financial interests.

**Change history:** A correction to this article has been published and is linked from the HTML version of this paper.

