## [Peer Review File · Nature Communications]

Reviewers' comments:

Reviewer #1 (Remarks to the Author):

The paper titled "Anisotropic scattering of hot electrons in an ultra-broadband plasmonic nanopatch metasurface" shows both a pump-probe measurements and theoretical calculation for determining different electron transport population and in particular the relationship with surface plasmons.

Three ingredients of the paper are skillfully combined together, such as colloidal silver nanocubes, pump&probe spectroscopy and LDA analysis.

The main conclusion regards that the study shows 3 different population of hot electrons. These conclusions are supported by a model, and the experimental measurement only partially contain this information. The study is detailed but only a structure, nanocubes of 500 nm side is used. I have doubts that this so big structure with a complex spatial and multipolar orders of hot-spot distribution, is the best choice to deduce the fast dynamical properties of hot electrons. Other geometries would be necessary to best elucidate the conclusions of this study, because hot electrons are strongly influence by the hot spot intensity and distribution.

Unfortunately, in my opinion, the paper doesn't show neither experimental nor theoretical breakthrough that justify the publication on Nature Communication. Moreover the style of the paper is most suitable for a more specialized journal.

Reviewer #2

The authors study the mechanism of hot-electron generation in plasmonic nanoantennas. The experimental results are very complete, including decay rate at different frequencies and from different contributions, and I think would be by themselves very interesting to the community. Furthermore, the authors propose very intriguing interpretations of these results, supported by a commendable theoretical analysis. While a definite understanding of hot electrons in plasmonic systems may still require further work, I believe this work is a significant advance towards this ambitious objective. The text is also generally well written, although sometimes I find the initial explanations are too brief (see below). I also describe below a couple of arguments that I had some problems in following, and some small technical questions, but these are relatively small points and do not retract to the importance of the paper. Thus, assuming the authors are able to reply satisfactorily to these questions, as I fully expect, I recommend this paper for publication in Nature Communications.

- Perhaps my main difficulty reading the paper is that some Figures were introduced very briefly (for example the schemes in Figure 1, Figure 2, Figure 4bc...). In most cases, a more complete explanation was given afterwards, so that it is possible to understand the work. Nonetheless, this configuration made it more difficult to me to understand some of the initial paragraphs as I was reading it for first time, and may make the text harder to follow for the general readership of Nature Communications.
- It could be useful to show in the main text (not just in SI) the simulated absorption spectra to compare with measurements
- In the discussion of the difference between 8 and 25nm, the authors seem to (qualitatively) compare the experimental decay rate with the theoretical spectra. How this is done was not clear to me from the text
- Sometimes the paper seems to focus on Au, sometimes on Ag, and I was not always very sure why. This was perhaps clearer in the discussion of Figure 5. I found particularly hard to follow the paragraph "Compared to the plasmon resonances" where I believe they are comparing two different sets of results but I am not sure to which figures they correspond.
- In general, they only mention briefly the figures in the SI. Thus, even if the SI contains much useful information to support the statements of the main text, I often did not realize until I read the SI (for example,

the detailed study of how patch antennas become a better candidate for hot-electron generation as the gap becomes narrow). Thus, it could help to point out more clearly in the main text what pieces of the arguments are supported by results in the SI.

- In the SI, the figures are occasionally mentioned but often not really described. The mentions are also often to the complete figure, and not to the separate panels. I think that referring more to the figures in the text of the SI could help the reader to better understand the arguments

The next comments are minor and are only included in case they may be useful to the authors. At least from my side, they can take the decision they see more fit and do not need to reply to each of them individually

- When the authors mention steady-state reflectivity measurements, they could point out that this corresponds to the purple line in Figure 2b.
- In the discussion in page 5, near “From them, it is apparent that ...” the authors focus on the fastest decay, but this was not initially clear to me, because other features can be more apparent (the mention to ultrafast could also refer to most of the components)
- When discussing the difference between the 8 and 25nm spacer, it is not clear which effect the changes on the resonance frequency can have on the measurements. This is discussed in more detail later, but at this stage it was not clear to me.
- Figure 3c-d are only briefly mentioned, but not really described what they show
- The explanation of Fig. 5ab is generally clear, but the authors do not discuss what they want to indicate by the arrows
- Near the end, in page 11, in “This can give the impression of a wavelength-dependence” do the authors mean “lack of wavelength dependence”?
- From the description of the simulations, I understand they are illuminating from the top. This illumination does not couple with some of the gap modes. I was just wondering if considering these modes –comparing the cases where they are and they are not exciting- could give extra information
- The authors use atomic units in (at least some of) the equations. It could be helpful to mention this
- In the caption of Figure 2c, I think “20x” refer to electric field enhancement E , but as the mention “intensity” it could also be understood as the enhancement of E^2
- In the caption of Figure 4c “The small peaks at 1100nm are artifacts from residual pump scatter” There are two rather large peaks around this wavelength (red, blue lines). I am not sure if the authors are also referring to these peaks or only to the smaller ‘fluctuations’
- The caption of Figure 6 could be made larger, and the e-e labels somewhat larger

Supplementary Material

- After equation S4, there is a strange format issue, at least in my copy: “however, nonthermal” appears at a lower height than the rest of the sentence
- In equation S11, I did not understand how S_{exp} is obtained from the experimental data
- In equations S11, S13, there appear the superscript “+” in the matrix. Is this the transpose, the inverse, a different matrix without superscript?
- In page 11, I did not understand the first sentences of the paragraph starting “In the gap mode, we observe...”

- In the caption of Figure S5, it was not initially clear to me what “percent absorbance in Ag” means (it could be understood as percent of the total incoming energy).
- In the captions of the figures, particularly in the SI, it may help the reader to make sure to always clarify which results are experimental and which one are theoretical, as this was occasionally not obvious to me.

Response to Reviewers

We thank the reviewers for their comments. We believe their comments and suggestions have helped us to substantially improve the manuscript and its impact. We address the comments point-by-point below. We note here that we chose to provide a slightly modified title, which is designed to increase impact by making the point that enhanced generation of hot electrons is a key discovery reported in this manuscript. In order to keep the title concise, we used “Coulomb scattering” as a substitute for “electron-electron” scattering, which has precedence in the literature (reference 14). In general, the manuscript is significantly revised throughout, and we feel it is greatly improved in clarity.

Reviewer 1

The paper titled "Anisotropic scattering of hot electrons in an ultra-broadband plasmonic nanopatch metasurface" shows both a pump-probe measurements and theoretical calculation for determining different electron transport population and in particular the relationship with surface plasmons. Three ingredients of the paper are skillfully combined together, such as colloidal silver nanocubes, pump&probe spectroscopy and LDA analysis.

We thank the reviewer for the positive comments.

The main conclusion regards that the study shows 3 different population of hot electrons. These conclusions are supported by a model, and the experimental measurement only partially contain this information.

We believe that the use of a metasurface to create absorption features throughout the UV-VIS-NIR spectral region in order to visualize hot electron kinetics at different wavelengths, and therefore at multiple symmetry points, is a significant advance. When combined with the fact that 3 decay rates were observed, the kinetic and spectral data strongly support the existence of 3 nonthermal hot electron decay pathways in silver and gold. The use of a bimetallic structures enables valuable spectral evidence for 2 of the 3 symmetry points where anisotropic scattering can occur (the 3rd symmetry point is outside of our equipment’s measurement range). When combined with the kinetics, strong support for our hypothesis of anisotropic scattering is provided. In response to the reviewer’s comment, we strove to improve the presentation in order to make these points clearer. Specifically, we now have a new section entitled “Signatures of anisotropic behavior.” The text in this section is also reworded to state that the Au spectral signatures serve as a control to further support the three different populations of hot electrons found in the kinetics and LDA analysis.

The study is detailed but only a structure, nanocubes of 500 nm side is used. I have doubts that this so big structure with a complex spatial and multipolar orders of hot-spot distribution, is the best choice to deduce the fast dynamical properties of hot electrons.

The nanocubes are actually 150nm on a side rather than 500nm. We believe the reviewer may have been observing the scale bar in the SEM image in Figure 1, which is 500 nm. The edge length was given in the schematic of the metasurface in that figure. In retrospect, the nanoparticle edge length was not specifically stated elsewhere in the main text of the manuscript (it was stated in the Methods Section), so we modified the text in the main manuscript in two places:

Figure 1 caption: Nanopatch metasurfaces were fabricated by depositing 150 nm (edge length) PVP-coated colloidal silver nanocubes on a 50 nm thick gold film supporting a thin Al₂O₃ spacer and interrogated with transient absorption spectroscopy.

Last paragraph before the results section: The metasurfaces employ silver nanocubes with 150 nm edge lengths that are separated from an underlying gold film by a thin polyvinylpyrrolidone (PVP) shell, a poly(allylamine hydrochloride) (PAH) adhesion layer, and an atomic layer deposition (ALD) grown Al₂O₃ spacer layer, creating an ensemble of nanopatch antennas (Figs. 1c,d and S2).

We also note other reasons why the 150nm cubes were focused on: (1) the larger structures produce a red-shifted absorption gap plasmon mode that is well separated from other modes, thereby enabling improved clarity in the analysis; (2) coherence of the plasmon improves at longer wavelengths due to decreased contributions from lossy interband transitions of bound electrons; (3) the spectral shifting of the gap mode as a function of gap thickness (such as shown in Figure 2 and Figure S5) from 900 nm to 1200nm remains entirely in the near infrared where an InGaAs CCD can be used for all gaps. By shifting to smaller particles (and therefore bluer wavelengths), needless technical difficulties and potential errors arise with changing between silicon and InGaAs CCDs.

Other geometries would be necessary to best elucidate the conclusions of this study, because hot electrons are strongly influence by the hot spot intensity and distribution.

This is an important comment that we worked hard to address. First, we added a new figure (Figure 4) which charts the expected nonthermal “hot” electron production for 10 different nanostructures under ultrafast optical excitation. As can be seen from the chart, the gap mode cube structures are superior to other structures involving isolated nanoparticles of varying shapes and composition. The chart also shows that smaller gaps are effective at producing more nonthermal electrons than larger gaps as we observe experimentally. The approach to the calculations is described in significant new text and equations (approximately two pages of text) in the new “Hot electron production” section beginning at the end of page 5.

Second, we emphasize that previous studies of ultrafast hot electron generation have been performed in a range of nanoparticle geometries as described for example in references 15, 17,

and 27-29. Only the gap mode structure that we first reported in reference 10 produces enough hot electrons to distinguish them from thermalized electrons as a rise and fall in ultrafast transient absorption/reflectivity experiments. Those studies were in all gold nanostructures, which as the calculations summarized in Figure 4 show, are not as effective as using Ag nanocubes. This is due to greater Drude damping in Au vs Ag, so that the enhanced electromagnetic fields produced by the plasmon are greater for Ag than Au.

To summarize, literature results combined with the new calculations and chart in Figure 4 strongly support the use of a metasurface that supports a gap plasmon mode to create an unusually large population of nonthermal carriers, particularly when Ag nanocubes are used.

Finally, in light of the reviewer's comment on the main conclusion, we have altered the title to read: "Enhanced generation and anisotropic Coulomb scattering of hot electrons in an ultra-broadband plasmonic nanopatch metasurface." This because we wish to emphasize that the main conclusion of the paper is twofold. The first is that these structures produce many nonthermal carriers with stronger time-resolved optical signals than previously observed in the literature. The second is that the nonthermal hot electrons undergo anisotropic electron-electron scattering. In order to keep the title concise, we used "Coulomb scattering" as a substitute for "electron-electron" scattering, which has precedence in the literature (reference 14).

Reviewer 2

The authors study the mechanism of hot-electron generation in plasmonic nanoantennas. The experimental results are very complete, including decay rate at different frequencies and from different contributions, and I think would be by themselves very interesting to the community. Furthermore, the authors propose very intriguing interpretations of these results, supported by a commendable theoretical analysis. While a definite understanding of hot electrons in plasmonic systems may still require further work, I believe this work is a significant advance towards this ambitious objective. The text is also generally well written, although sometimes I find the initial explanations are too brief (see below). I also describe below a couple of arguments that I had some problems in following, and some small technical questions, but these are relatively small points and do not retract to the importance of the paper. Thus, assuming the authors are able to reply satisfactorily to these questions, as I fully expect, I recommend this paper for publication in Nature Communications.

We thank the reviewer for the positive comments.

Perhaps my main difficulty reading the paper is that some Figures were introduced very briefly (for example the schemes in Figure 1, Figure 2, Figure 4bc...). In most cases, a more complete explanation was given afterwards, so that it is possible to understand the work. Nonetheless, this configuration made it more difficult to me to understand some of the initial

paragraphs as I was reading it for first time, and may make the text harder to follow for the general readership of Nature Communications.

We agree with the reviewer that some of the descriptions of Figures were brief and could be improved. We revised the text, especially in the opening paragraphs to describe the Figures in improved detail. For example, the manuscript previously simply referred to “Fig. 1” initially, whereas now we refer to Fig. 1a or Fig. 1b, etcetera in our initial descriptions of the schemes.

We also believe that our efforts to more completely explain the motivations for pursuing a metasurface for gap plasmon generation through the addition of Figure 4 and the surrounding text also substantially improve the flow of the manuscript and detail key concepts of nonthermal generation earlier. This is contained in the section now called “hot electron production” Equations 1 and 2 and surrounding text are new and are designed to more completely introduce nonthermal hot electron generation in plasmonic nanostructures. Overall, this means that we have added introductory material and text, such that now the previous Figure 2 which contains the main time-resolved experiments and complementary electromagnetic simulations is now Figure 5. Figure 2 and 4 are new figures, and Figure 3 is earlier in the manuscript to more thoroughly introduce our approach and concepts of nonthermal hot electron generation before reaching Figure 5.

It could be useful to show in the main text (not just in SI) the simulated absorption spectra to compare with measurements

We agree with the reviewer. A new Figure 2 has been added to include simulated absorption spectra for the two spacer thicknesses (8nm and 25nm) most discussed in the manuscript. It also now includes not only the total simulated absorption, but a breakdown of the absorption contributions from the Ag nanocubes and the Au film, which is designed to complement the individual components discussed in Figures 3c and 3d on nonthermal generation rates and population, respectively. This is designed to be additional useful information to the reader to better understand how each component contributes to the overall nonthermal generation rate. The SI figure S5 also contains the simulated spectra and individual components for all spacer thicknesses used.

In the discussion of the difference between 8 and 25nm, the authors seem to (qualitatively) compare the experimental decay rate with the theoretical spectra. How this is done was not clear to me from the text

We agree with the reviewer that this presentation could have been more clear. As described in the response to reviewer 1, we added considerably more detail to the manuscript on the calculations that produce the comparisons made in Figure 3. These paragraphs are added to the new “hot electron production” section beginning with equations 1 and 2 and surrounding text. We believe this additional detail substantially augments Figure 3, which shows that both experimental and theory show that smaller gaps produce many more nonthermal electrons,

and that the gap mode strongly enhancing nonthermal electron production relative to SiO₂, where the kinetic response shows no ultrafast nonthermal electron component.

Sometimes the paper seems to focus on Au, sometimes on Ag, and I was not always very sure why. This was perhaps clearer in the discussion of Figure 5. I found particularly hard to follow the paragraph “Compared to the plasmon resonances” where I believe they are comparing two different sets of results but I am not sure to which figures they correspond.

This is a reasonable comment. We believe that the changes we made to the manuscript involving Figures 2, 3, and 4 described above begin to address this comment because they include the impact of the individual Ag and Au components on optical absorption and nonthermal hot electron generation. We have also added headings throughout the manuscript to assist the reader in following the manuscript and to be consistent with Nature Communications guidelines. This text is now part of the “Signatures for anisotropic behavior” section where we explain the three decay kinetics observed for nonthermal hot electron decay. Finally, the reason for probing Au is essentially as a control experiment for the observations made for Ag. This is because probing Ag gives us access to the plasmonic modes which produce the strongest responses from nonthermal hot electrons, but Au provides valuable complementary time-resolved spectra to support our proposal for anisotropic decay. We do agree with the reviewer that this connection for probing nonthermal electrons in Au was not made clear enough. Thus we have added the following two statements:

Page 12: Until now, we have focused our analysis on the three plasmon resonances of the metasurface, which indiscriminately couple to intraband transitions in the Ag nanocubes (blue arrows in Fig. 7a). As a control experiment in support of our observations for Ag nanocubes, we now consider the response of the gold IB transition,

Page 13: We find the slow nonthermal $e-e$ scattering time to be ~37% longer in gold, consistent with the predicted value of 35%.¹⁴ Thus the combination of observing 3 distinct population decays in Au that are slower than Ag by the predicted value, combined with the additional spectral evidence made available only in Au, is strong support for anisotropic nonthermal hot electron decay in these metasurfaces.

In general, they only mention briefly the figures in the SI. Thus, even if the SI contains much useful information to support the statements of the main text, I often did not realize until I read the SI (for example, the detailed study of how patch antennas become a better candidate for hot-electron generation as the gap becomes narrow). Thus, it could help to point out more clearly in the main text what pieces of the arguments are supported by results in the SI.

We have now broken the SI into sections (Supplementary Notes 1-7) which we now individually reference within the main text, along with relevant figures in the SI.

In the SI, the figures are occasionally mentioned but often not really described. The mentions are also often to the complete figure, and not to the separate panels. I think that referring more to the figures in the text of the SI could help the reader to better understand the arguments

We agree with the reviewer that more detail could have been given in the Supplementary Information, particularly when referring to figures. The SI is now broken into Supplementary Notes sections numbered 1-7 and all of the figures in the SI are referred to either in the SI or in the main text. We also added more specificity regarding which figure panel we referred to where it was appropriate.

The next comments are minor and are only included in case they may be useful to the authors. At least from my side, they can take the decision they see more fit and do not need to reply to each of them individually

When the authors mention steady-state reflectivity measurements, they could point out that this corresponds to the purple line in Figure 2b.

We have clarified in the in the figure caption of Figure 5 (formerly Figure 2) that the purple curve in panel 5b corresponds to the steady-state spectrum. We have also reorganized our discussion of the steady-state spectra into a new section titled “Optical characterization of the metasurface” at the beginning of the results section. This highlights the spectra of the new Figure 2 and details the multiple resonances present in the sample.

In the discussion in page 5, near “From them, it is apparent that ...” the authors focus on the fastest decay, but this was not initially clear to me, because other features can be more apparent (the mention to ultrafast could also refer to most of the components)

We have further clarified this text by adding “From them, it is apparent that the strongest ultrafast (< 300 fs) response...” to indicate we’re referencing the fastest decay.

When discussing the difference between the 8 and 25nm spacer, it is not clear which effect the changes on the resonance frequency can have on the measurements. This is discussed in more detail later, but at this stage it was not clear to me.

We believe the new Figure 4, which specifically describes the impact of the 8 and 25 nm spacer thicknesses on the rate of nonthermal electron generation addresses this comment. We also believe that the new Figure 2, which shows the experimental and calculated spectra for the 8 and 25 nm spacers (including the individual Ag and Au components)also contributes to the readers’ understanding of the differences between the different thicknesses

Figure 3c-d are only briefly mentioned, but not really described what they show

This is now addressed by the new theory section in the main text in the “hot electron

production” section which specifically refers to Figure 3c-d.

The explanation of Fig. 5ab is generally clear, but the authors do not discuss what they want to indicate by the arrows

This is now Figure 7a, b. We agree we could have been clearer. We re-worded the figure caption for Figure 7 to rigorously define the blue and red arrows. We also revised the text to be clearer – please see the paragraph beginning with “Until now...”

Near the end, in page 11, in “This can give the impression of a wavelength-dependence” do the authors mean “lack of wavelength dependence”?

We revised this sentence for clarity to read: “This can incorrectly give the impression of wavelength dependent e - e scattering rates when in fact, a global analysis reveals that different wavelengths measure different percentages of contributions from nonthermal carriers.”

From the description of the simulations, I understand they are illuminating from the top. This illumination does not couple with some of the gap modes. I was just wondering if considering these modes –comparing the cases where they are and they are not exciting- could give extra information

The angular dependence of the metasurface reflectivity has been extensively characterized previously (see Ref. 15) and shown little to no angular dependence of the modes. Due to the subwavelength size of the nanocubes, light effectively couples to the gap mode directly from free space. Our electromagnetic simulations do not indicate other (dark) modes within the configurations that we are considering.

The authors use atomic units in (at least some of) the equations. It could be helpful to mention this

The a.u. notation was meant to denote “arbitrary units.” We have changed the notation on the figures and equations to read “arb. u.” to be more clear.

In the caption of Figure 2c, I think “20x” refer to electric field enhancement E , but as the mention “intensity” it could also be understood as the enhancement of E^2

We apologize for the confusion and have corrected the figure caption to read “field strength” as opposed to intensity.

In the caption of Figure 4c “The small peaks at 1100nm are artifacts from residual pump scatter” There are two rather large peaks around this wavelength (red, blue lines). I am not sure if the authors are also referring to these peaks or only to the smaller ‘fluctuations’

For clarity we have corrected this to read “The small fluctuations at 1100 nm...” as the reviewer

has suggested.

The caption of Figure 6 could be made larger, and the e-e labels somewhat larger

We have made the labels in the figure (now Figure 8) larger and added additional text to the figure caption to better explain the importance of the data to our proposed mechanism of anisotropic nonthermal electron decay.

Supplementary Material

After equation S4, there is a strange format issue, at least in my copy: “however, nonthermal” appears at a lower height than the rest of the sentence

We thank the reviewer for pointing out this formatting issue and have now resolved it.

In equation S11, I did not understand how S_{exp} is obtained from the experimental data

S_{exp} refers to the time-dependent experimental data. We have updated the preceding sentence to better reflect this: “The amplitudes for each component are then found by linear fitting the product of the pseudoinverse of the guess matrix and the experimental data (S_{exp}) for a given wavelength...”

In equations S11, S13, there appear the superscript “+” in the matrix. Is this the transpose, the inverse, a different matrix without superscript?

The “+” superscript refers to the pseudoinverse, which we have now explicitly defined in the text.

In page 11, I did not understand the first sentences of the paragraph starting “In the gap mode, we observe...”

We have rewritten the opening sentences of this paragraph to be more clear.

In the caption of Figure S5, it was not initially clear to me what “percent absorbance in Ag” means(it could be understood as percent of the total incoming energy).

We have corrected the caption to read “...percent of light absorbed in Ag...” as it refers to the percent of total absorbed energy being absorbed within the Ag nanocubes. Note that this is now Figure S6.

In the captions of the figures, particularly in the SI, it may help the reader to make sure to always clarify which results are experimental and which one are theoretical, as this was occasionally not obvious to me.

We have updated the figure titles/captions to more clearly differentiate what data was simulated and measured.

REVIEWERS' COMMENTS:

Reviewer #1 (Remarks to the Author):

I think that the authors improved significantly the paper in several aspects, clarity and experimental completeness. I agree that the paper can be published now in the present form. Due to the fact that up to now very few experimental papers are published on hot electrons nanoscopy, that I consider strictly related to the present paper, I would suggest to include in the references the following papers:

1) Hot-electron nanoscopy using adiabatic compression of surface plasmons, A Giugni et al., Nature nanotechnology 8 (11), 845-852, 2013

2) Experimental Route to Scanning Probe Hot Electron Nanoscopy (HENs) Applied to 2D Material, A Giugni et al., Advanced Optical Materials DOI: 10.1002/adom.201700195 8 June 2017

Reviewer #2

The authors have satisfactorily answered my questions, and further strengthened their paper. The combination of experiments, signal analysis and calculations is particularly attractive; although it may not be a full proof, it gives significant support to the theoretical claims in the paper. I thus think that this contribution is an important contribution to the topic of hot-electron generation, which will stimulate further discussion and experiments, and recommend this paper for publication.

I just add below a few comments on small points

In line 183 "We observe four differential absorption regimes" it was not very clear to me what 'regime' meant in this context

Have the authors study the fluence dependence of the e-ph thermal carriers, to see if it supports their conclusions?

I am possibly missing a trivial factor, or the authors may be using a different definition of dephasing than I am used to, but if I convert 13 fs into a linewidth I obtain $\hbar \cdot 1/\tau_{\text{echarge}} = 50 \text{ meV}$.

In Figure 5b, the label "A" for the "Steady-state reflectivity" is not intuitive.

In the SI. Lines 38-40 were not very clear to me. In particular, when comparing the structure, they seem to go from almost zero (near 100% in silver) to 70% absorption in gold, but the latter is qualified as 'only'. Furthermore, Figure S7 indicates strong absorption in gold. Do the authors meant that for the multipolar plasmon resonance near "100% nonthermal carrier generation occurs in gold"?

In the SI, line 266, do "a decrease of nonthermal carrier" refer to the absolute value or to the fraction?

Reviewer #1

I think that the authors improved significantly the paper in several aspects, clarity and experimental completeness. I agree that the paper can be published now in the present form.

We thank the reviewer for their work and are pleased that they find the paper improved.

Due to the fact that up to now very few experimental papers are published on hot electrons nanoscopy, that I consider strictly related to the present paper, I would suggest to include in the references the following papers:

1) Hot-electron nanoscopy using adiabatic compression of surface plasmons, A Giugni et al., Nature nanotechnology 8 (11), 845-852, 2013

2) Experimental Route to Scanning Probe Hot Electron Nanoscopy (HENs) Applied to 2D Material A Giugni et al., Advanced Optical Materials DOI: 10.1002/adom.201700195 8 June 2017

We agree that the above papers are directly related to this study and have added the suggested papers as references 4 and 5 of the main text. Additionally, we have updated the introductory paragraph with:

“Hot electrons have been demonstrated to inject over large interfacial energy barriers, enabling sensitization of plasmonic Schottky photodetectors to sub-bandgap photons,¹⁻³ nanoscopy with high spatial and chemical sensitivity,^{4,5} and photocatalyzed reactions including hydrogen dissociation on plasmonic nanoparticles.⁶⁻⁹”

Reviewer #2

The authors have satisfactorily answered my questions, and further strengthened their paper. The combination of experiments, signal analysis and calculations is particularly attractive; although it may not be a full proof, it gives significant support to the theoretical claims in the paper. I thus think that this contribution is an important contribution to the topic of hot-electron generation, which will stimulate further discussion and experiments, and recommend this paper for publication

We thank the reviewer for their comments.

In line 183 “We observe four differential absorption regimes” it was not very clear to me what ‘regime’ meant in this context

We apologize for the confusion and have changed “regimes” to “features” for improved clarity.

Have the authors study the fluence dependence of the e-ph thermal carriers, to see if it supports their conclusions?

We chose to focus this work on the e-e scattering rates, as the e-ph scattering of thermal carriers has been well-documented in the literature (see Refs. 17-19, 29-31). As such, a detailed discussion on the fluence-dependent e-ph dynamics are outside the scope of this study. However, we have added the following line to the final paragraph of Supplementary Note 2:

”However, we do observe a decreased e-ph scattering rate with increasing fluence.”

I am possibly missing a trivial factor, or the authors may be using a different definition of dephasing than I am used to, but if I convert 13 fs into a linewidth I obtain $\hbar/\tau_{\text{e-ph}} = 50 \text{ meV}$.

To convert between dephasing time (T_2) and the plasmon resonance linewidth (FWHM in eV), the equation to be used is $T_2 = 2\hbar/\text{FWHM}$. This yields the values reported in the main text.

In Figure 5b, the label “A” for the “Steady-state reflectivity” is not intuitive.

We apologize for the confusion and have updated the figure caption to read “Steady-state absorbance”.

In the SI. Lines 38-40 were not very clear to me. In particular, when comparing the structure, they seem to go from almost zero (near 100% in silver) to 70% absorption in gold, but the latter is qualified as ‘only’. Furthermore, Figure S7 indicates strong absorption in gold. Do the authors meant that for the multipolar plasmon resonance near “100% nonthermal carrier generation occurs in gold”?

We agree with the reviewer that the previous wording was unclear and have updated this sentence to read:

“Our calculations indicate nearly 100% of nonthermal carrier generation occurs in the Ag nanocubes when exciting at the multipolar plasmon resonance (at ~370 nm), while 55% and 70% are generated in Au at the gap and quadrupolar plasmon resonances (at ~1000 nm and 630 nm), respectively.”

This is highlighting the fact that while there may be strong absorption in gold (e.g. at the multipolar resonance), this does not necessarily translate to a high nonthermal electron generation rate. At the multipolar resonance, the nonthermal generation is dominated by the nanocubes, while it is more evenly distributed at the other two plasmon resonances.

In the SI, line 266, do “a decrease of nonthermal carrier” refer to the absolute value or to the fraction?

We have corrected this line to read “a decreased contribution from nonthermal carriers” for clarity.